# Fungal Diversity and Composition of the Continental Solar Saltern in Añana Salt Valley (Spain)

**DOI:** 10.3390/jof7121074

**Published:** 2021-12-14

**Authors:** Maia Azpiazu-Muniozguren, Alba Perez, Aitor Rementeria, Irati Martinez-Malaxetxebarria, Rodrigo Alonso, Lorena Laorden, Javier Gamboa, Joseba Bikandi, Javier Garaizar, Ilargi Martinez-Ballesteros

**Affiliations:** 1MikroIker Research Group, Department of Immunology, Microbiology and Parasitology, Faculty of Pharmacy, University of the Basque Country UPV/EHU, Paseo de la Universidad 7, 01006 Vitoria-Gasteiz, Spain; maiaazpiazu29@gmail.com (M.A.-M.); albadaaira@gmail.com (A.P.); irati.martinez@ehu.eus (I.M.-M.); rodrigo.alonso@ehu.eus (R.A.); lorena.laorden@ehu.eus (L.L.); joseba.bikandi@ehu.eus (J.B.); javier.garaizar@ehu.eus (J.G.); 2Department of Immunology, Microbiology and Parasitology, Faculty of Science and Technology, University of the Basque Country UPV/EHU, Barrio Sarriena s/n, 48940 Leioa, Spain; aitor.rementeria@ehu.eus; 3Biogenetics, Portal de Zurbano 3, 6-B, 01013 Vitoria-Gasteiz, Spain; jgamboa@biogenetics.es

**Keywords:** ITS, metabarcoding, fungi, biodiversity, continental saltern

## Abstract

The Añana Salt Valley in Spain is an active continental solar saltern formed 220 million years ago. To date, no fungal genomic studies of continental salterns have been published, although DNA metabarcoding has recently expanded researchers’ ability to study microbial community structures. Accordingly, the aim of this present study was to evaluate fungal diversity using the internal transcribed spacer (ITS) metabarcoding at different locations along the saltern (springs, ponds, and groundwater) to describe the fungal community of this saline environment. A total of 380 fungal genera were detected. The ubiquity of *Saccharomyces* was observed in the saltern, although other halotolerant and halophilic fungi like *Wallemia*, *Cladosporium*, and *Trimmatostroma* were also detected. Most of the fungi observed in the saltern were saprotrophs. The fungal distribution appeared to be influenced by surrounding conditions, such as the plant and soil contact, cereal fields, and vineyards of this agricultural region.

## 1. Introduction

Saline and hypersaline environments are widely distributed around the world. The study of these habitats is becoming more common, not only as a means to increase knowledge regarding the structure of their microbial communities but also to gain insights into the adaptations of these organisms to these unique locations. Among hypersaline environments, solar salterns are of particular interest. Different types of salterns exist that may be classified as littoral salterns and continental or inland salterns. Continental salterns are those that use brine obtained from spring sources, for the production of salt via evaporation and solar activity [1].

The Añana Salt Valley (30 km southwest of Vitoria-Gasteiz, Álava, Northern Spain; 42.80 N 2.98 W) is an active continental solar saltern formed about 220 million years ago following the evaporation of water from the great ocean (Tethys Ocean) that covered most of the Earth. This process led to the deposit of extensive layers of evaporites (salts, anhydrite, and gypsum), which, emanating from highly plastic stratigraphic levels and subjected to great pressure, rose through the sedimentary layers of the Earth’s crust, crossing and deforming these layers and resulting in the formation of a diapir. Rainwater crosses the upper stratum of the diapir rock, and then layers of salt appear on the surface decades later in the form of hypersaline or brackish springs [2]. Depending on the path the filtrated water takes through the rocks, it encounters areas of higher or lower salt content. For this reason, springs that are next to each other contain water with different levels of salinity. 

The hypersaline springs located in the Añana Salt Valley are water sources that supply brine at the surface level in a natural and continuous way without any need for drilling or for the use of pumps. Archaeological remains show that these water sources have been used by locals for 7000 years, for the extraction of salt [3,4]. Consequently, the physical, chemical, and anthropogenic characteristics of the Añana Salt Valley environment make it an exceptional setting for study and research. What is more, these salt flats contain deposits with important paleo-environmental and paleo-climatic information, in addition to the biodiversity of their extreme environments [3,4].

For many years, saline and hypersaline environments were thought to be populated almost exclusively by prokaryotic microorganisms. Recently, it has been shown that eukaryotic microorganisms are present too [5,6,7]. Studies regarding prokaryotic diversity and distribution have been conducted at different salterns using both culture-dependent and culture-independent methods [8,9,10,11,12]. Fewer studies have been conducted on the examination of fungal communities [13,14,15]. In the last few years, DNA metabarcoding has become an effective tool for the identification of yeast and other fungal species. Different studies have been conducted on soil, marine, and saline ecosystems; vineyards and wine-production systems; and in glaciers [16,17,18,19,20,21,22,23], among other ecosystems. Although a study on prokaryotic and virus diversity in the Añana Salt Valley has been published [24], no fungal metabarcoding survey in this ecosystem has been published nor of other continental or inland salterns worldwide. The aim of this present study, therefore, was to evaluate fungal diversity in the extreme ecosystem of the Añana Salt Valley using metabarcoding to describe the composition of the fungi present in this location.

## 2. Materials and Methods

### 2.1. Sampling Sites and Proceedings

Water samples from diverse sites around the Añana Salt Valley were collected to perform the fungal diversity study (Figure 1). Individual water samples from each of the seven sites were collected during the survey and studied for fungal diversity. Two of these sites, Santa Engracia (SE) Spring and Pico Spring, were springs containing high-salinity water. SE Spring is the main brine supplier for salt production in the saltern, with an average flow of 3 L/s and 200 g of salt/L. Pico Spring provides a lower brine flow to the system. The salt production brine is distributed by an interconnected wooden canal system, channeling the brine to a series of distribution ponds and crystallizing pans. Three of these ponds (Pond I, II, and III), containing high-salinity water and located along the salt-production system, were also selected for this study. In the valley, other springs with a lower-salinity water content arise naturally. From among these, brackish water from the Pico Dulce Spring was analyzed. Finally, brackish groundwater from an aquifer at 60 m depth, accessible through a piezometer called S8, was also analyzed. Water samples were collected in October 2017 (SE, Pico Spring, and ponds) and in June 2018 (S8 and Pico Dulce Spring).

Water samples were directly obtained using sterile glass bottles or a Niskin bottle (Aquatic BioTechnology, El Puerto de Santa María, Spain). Groundwater from the S8 piezometer was taken at a 60 m depth using a manual bailer system (Eijkelkamp, Giesbeek, The Nederlands). Physicochemical parameters, such as water temperature, pH, NaCl concentration, and conductivity (related to salt content), were determined in situ (Combo tester, Hanna Instruments, Eibar, Spain). Water samples were taken directly to the laboratory for processing. 

### 2.2. DNA Extraction and Sequencing

Samples were enriched by filtering approximately 5 L of water in aseptic conditions through a tailored device, driven by vacuum/compression pumps, which concentrated the samples via a series of decreasing flow-rate polycarbonate membranes (Labbox, Premia de Dalt, Spain), ending in a 0.2 μm pore size filter. DNA extraction and quantification of the samples were conducted according to the manufacturer’s instructions via the Genomic Mini AX Yeast Kit (A&A Biotechnology, Gdansk, Poland) and the QuantiFluor dsDNA System (Promega, Madison, WI, USA), respectively. The fungal internal transcribed spacer 1 (ITS1) region was amplified, as suggested in the ITS Metagenomics Protocol of Illumina by modified ITS1-F and ITS2 primers set [25]. The library was prepared by a Nextera DNA Library prep kit (Ilumina, San Diego, CA, USA) according to that protocol. High-throughput sequencing (HTS) was performed on an Illumina MiSeq platform, which generates paired-end sequences in FASTQ format. The nucleotide sequence data from the study are available in the DDBJ/EMBL/GenBank databases under the accession number PRJNA749727.

### 2.3. Sequencing Data and Statistical Analysis

Most of the analyses was carried out using QIIME2 version 2021.2 software tools and pipelines [26]. Joining of paired-end reads, sequence quality control, and feature table construction were performed by denoising with DADA2 plugin (q2-dada2) [27]. During this step, sequence denoising, dereplication, and chimera filtering were undertaken. Sequences were trimmed at a 200 bp length. Amplicon sequence variants (ASVs) [28] generated by DADA2 were assigned taxonomically using the q2-feature-classifier [29] classify-sklearn naïve Bayes taxonomy classifier with the UNITE 8.3 dynamic database [30] at 97% similarity level. Manually searching of unidentified ASVs was performed by BLAST (https://blast.ncbi.nlm.nih.gov/Blast.cgi, accessed on 6 October 2021). 

Alpha and beta diversity metrics were estimated using q2-diversity plunging after rarefaction of the samples to 2270 sequences per sample. Alpha diversity estimators, such as observed ASVs and Chao1 index, were determined to establish the species richness at each point. To study the diversity of the species in the samples, α estimators, such as Shannon and Simpson indexes, were used: the higher the Shannon index, the greater the diversity, and the closer the Simpson index is to 0, the lower the diversity. Pielou’s evenness, which quantifies how equal a community is, was also determined (the closer to 0, the higher the dominance of a species). Statistical significance of the alpha diversity values between sampling sites was assessed using the Kruskal–Wallis H test. A *p* value < 0.05 was considered statistically significant. To establish the correlation of diversity indexes with physicochemical characteristics, Pearson’s coefficient test was performed (*p* value < 0.05 was considered statistically significant). Dissimilarity between sample groups was also examined by the Bray–Curtis coefficient and differences among groups were assessed by ANOSIM with 999 permutations (*p* < 0.05). A heatmap was performed by ggplot2 R package to visualize differences at the genus level composition among samples. Associations between samples were established by the Bray–Curtis dissimilarity. Venn analysis was performed by Venn diagram software (available online at: http://bioinformatics.psb.ugent.be/webtools/Venn/, accessed on 21 July 2021) to determine common and unique fungal taxa on samples. 

The FUNGuild database was used to assign ecological guilds within three trophic modes (i.e., pathotroph, symbiotroph, and saprotroph) to all ASVs [31]. Only fungal ASVs with probable and highly probable confidence matches were considered for further analysis. 

## 3. Results

### 3.1. Water Physicochemical Data

The physicochemical characteristics of water samples measured in situ are shown in Appendix A. Two different degrees of salinity were observed, which were used to establish groups for further analysis: SE Spring, Pico Spring, and the three brine ponds were classified as salty water, and Pico Dulce Spring and S8 piezometer were classified as brackish water due to their lower salt concentration.

### 3.2. Sequence Analysis and ASV Determination

The fungal diversity and community structure in the water samples taken from representative sites in the Añana Salt Valley were investigated by HTS (Figure 1). After denoising and quality control of sequenced DNA, a mean of 132,520 reads per sample was obtained. A total of 704,874 sequences were obtained corresponding to 2204 different ASVs determined by DADA2. These ASVs were subsequently assigned at different taxonomy levels for fungal identification. Among the ASVs, 853 failed to be identified at any known phylum (38.7% of the total). 

### 3.3. Fungal Diversity in the Saltern

Rarefaction curves were computed by rarefying each sample to the minimum number of sequences (2270 reads). Rarefaction curves were not parallel for all the samples analyzed but started to flatten (Appendix A). The alpha diversity was calculated after normalization of the samples. Table 1 shows the taxa richness and diversity values obtained by the different alpha diversity estimators. SE Spring had the lowest diversity. There, the smallest number of ASVs (71 ASVs) was detected and the Shannon index was 3.59 (ranging from 4.89–6.47 for the rest of the locations). The other six locations demonstrated a similar diversity, except for S8, which was less diverse. While 93 ASVs and a Shannon index of 4.89 were obtained at S8, the observed ASVs from the other five locations ranged from 262–396 and the Shannon index ranged from 5.44–6.47. Pielou’s evenness of SE was 0.58 (the closest value to 0 in the study), which indicated the presence of a dominant taxon (presumably *Saccharomyces*). No statistically significant differences were observed in the alpha diversity indexes between sample sites.

The diversity and pH showed a statistically significant correlation based on Pearson’s coefficient test (Table 2), indicating that environmental diversity decreases as the pH of the water decreases too. No correlation was observed in regard to the NaCl concentration or temperature.

The ANOSIM test based on Bray–Curtis dissimilarity identified differences in the fungal diversity among groups based on their salinity (salty and brackish water) (*p*-value 0.046). No differences were observed regarding different water types (spring, pond, or groundwater).

### 3.4. Fungal Taxonomic Assignment and Community Composition

The taxonomic assignment of ASVs showed that *Basidiomycota* and *Ascomycota* were the phyla present in the saltern (0.18–63.6% and 23–81.1%, respectively) (Figure 2). No other fungal phylum was detected in the samples. As referenced above, 38.7% of the ASVs were classified only as fungi. The manual identification of unidentified ASVs by BLAST search did not improve these results, so a low taxonomic assignment to the species level was obtained in the study. The taxonomic assignment of each ASV can be found in Appendix A. A total of 380 fungal genera were detected in the survey (Appendix A). In Figure 2, genera whose relative abundance was more than 3% in at least one of the locations analyzed at the saltern are shown. Regarding SE Spring, 30 ASVs were taxonomically assigned to genus level; 184 with respect to Pico Spring; 147, 196, and 110 genera with respect to Pond I, II, and III, respectively; 81 at Pico Dulce Spring; and 28 genera at S8. 

The relative abundance of the genera varied among the sampling sites. *Saccharomyces* was present at all locations; its relative abundance ranged from 1.5–49.9%. Several other genera were also detected at all of the sites, including *Hanseniaspora* (0.2–21.9%), *Mycosphaerella* (0.4–3.8%), *Vishniacozyma* (0.1–24.2%), and *Alternaria* (0.1–2.4%).

SE Spring was characterized principally by the presence of *Saccharomyces* (49.9%) and *Hanseniaspora* (21.9%). At Pico Spring, *Saccharomyces* was also detected (8.4%), followed by *Peniophora* as the second most abundant genus (4.3%). The three ponds analyzed along the salt-production system have several genera in common, but the fungal composition differed between them. Pond I was mainly comprised of *Wallemia*, *Filobasidium*, and *Vishniacozyma* genera (24%, 18.1%, and 11.2%, respectively). Pond II was comprised of *Filobasidium* and *Epicoccum* (10% and 7.5%), and at Pond III, *Saccharomyces* and *Epicoccum* were mainly detected (20.1% and 8%). The principal genus that made up the fungal community of the low-salinity Pico Dulce Spring water was *Vishniacozyma* (18.4%). This genus was also detected in groundwater from S8 piezometer (24.2%). *Pichia* was also detected (12.6%) as the second most abundant genus at this location.

In total, 617 ASVs were able to be identified to the species level (28% of the total ASVs). Thirty-seven different species were detected at SE Spring; 188 at Pico Spring; 129, 92, and 179 species at Pond I, II, and III, respectively; 54 at Pico Dulce Spring; and 20 at S8. Appendix A shows species whose relative abundance is higher than 0.5% at each location. *Pichia mandshurica*, *Cladosporium delicatulum*, and *Mycospharella tassiana* were the main species detected in brackish water samples. The presence of *C. delicatulum* and *M. tassiana* was also notable in the ponds. *Wallemia ichthyophaga* was the most abundant species detected at Pond I (24%) and *Epicoccum dendrobii* at Pond II and III (6.5% and 7.7%, respectively). *Vishniacozyma victoriae* was largely found also at Pond I and II (10.5% and 2.8%, respectively). Salty springs, SE Spring and Pico Spring, had different species compositions compared to the other sampling sites, and between them. At SE Spring, *Saccharomyces bayanus*, *Metsechnikowia shanxiensis*, and *Lachancea quebecensis* were the main species identified (3.4%, 2.7%, and 1.7%, respectively). *Stereum hirsutum* (1.2%), *Resupinatus europeus* (1.1%), and *Keissleriella rosarum* (1%) were found at Pico Spring.

The distribution of the most representative genera (>1% in at least one of the samples) among the sampling sites is shown in the heatmap (Figure 3). Although some genera were detected in all of the samples, different composition profiles at the genus level can be defined. The clustering performed by the Bray–Curtis dissimilarity coefficient based on the genera abundance showed that SE Spring differs from the rest of the sites analyzed, and that brackish water sites are more similar to each other than to other salty water locations (Figure 3). 

As mentioned above, five genera were common among all the samples analyzed: *Saccharomyces*, *Alternaria*, *Mycosphaerella*, *Hanseniaspora*, and *Vishniacozyma*. Unique fungal taxa were detected from each sample site, with the exception of S8. From SE Spring, 17 taxa were identified as exclusive to this location and 30 were present only at Pico Dulce Spring. The ponds showed the highest number of unique taxa, with 47, 118, and 31 taxa identified for Ponds I, II, and III, respectively. When analysis was carried out according to the water salinity of the locations, 89 common fungal taxa were identified for salty and brackish water types; however, 516 and 40 were unique to salty and brackish water, respectively (Figure 2).

### 3.5. Fungal Functional Guilds

FUNGuild was used to identify fungal ecological functions at each sampling site (Figure 4). A notably different fungal functional profile was identified at SE Spring. It was mainly composed of saprotrophs (95.5%), while the rest of the trophic modes represented a minor abundance or none. At Pico Spring and Pond I, II, and III, saprotrophs were also in the greatest abundance (73.1%, 85.4%, 54.2%, and 42.7%, respectively), although pathotrophs (5.2%, 5.7%, 16.8%, and 21.6%), pathotroph−saprotrophs (17.4%, 5.1% 6.0%, and 7.4%), and pathotroph−saprotroph−symbiotrophs (2.9%, 1.8%, 17.8%, and 20.8%) were also detected. In addition, symbiotrophs were detected only at those locations, albeit at a low-abundance percentage. By contrast, Pico Dulce Spring water and S8 groundwater were composed mainly of pathotroph-saprotophs (40.1% and 42.2%, respectively), followed by saprotrophs (34.3%) at Pico Dulce, and pathotroph-saprotroph-symbiotrophs (32.8%) in S8 groundwater. 

The rest of the trophic modes were present in less than 5%. The saprotrophs comprised mainly undefined saprotrophic fungi. Pathotrophs consisted of plant pathogens at all sampling sites apart from SE Spring, where animal pathogens were the main guild in this mode. Symbiotrophs were mainly dominated by lichenized and epiphytic fungi.

## 4. Discussion

Hypersaline environments are extreme habitats about which there is very little information. Most of the studies conducted in relation to such ecosystems have aimed to study prokaryotic microorganisms, although few surveys have considered fungal community studies as a part of their objectives. It is now known that eukaryotic microorganisms are also part of these unique ecosystems. Recent diversity surveys using sequencing approaches have demonstrated that fungi colonize a wide variety of aquatic ecosystems and that they possess unique adaptations methods for life in a saline-water environment [32,33]. For this present study, we aimed to determine and understand the fungal composition of the Añana Salt Valley continental solar saltern. For this purpose, various locations across the valley (spring water and groundwater) and ponds along the salt-production system (brine distribution and resting ponds) were studied using ITS metabarcoding analysis. Nonetheless, the presence of fungal DNA must be carefully interpreted as it could simply represent the casual presence in these habitats of dead cells or spores transmitted by insects, birds, plants, human activities, or the wind.

Fungal taxa identified at Añana Salt Valley were *Ascomycota* and *Basidiomycota*. In particular, *Ascomycota* members were more frequently detected than *Basidiomycota.* These results were also previously observed in deep ocean ecosystems [34], saline soils [35], deep-sea sediments [36,37], and in other salterns [14]. Furthermore, Wei et al. [15] have recently described the fungal composition of a millennial coastal saltern in South China, where the fungal community was different from that which was detected in the continental saltern of this study.

Among the *Ascomycota* phylum, *Saccharomyces* was the most abundant yeast detected in the saltern. This ubiquitous organism is widely distributed in different environments from natural to domesticated strains [38]. The capacity to adapt to saline conditions is also known in *Saccharomyces* strains [39]; therefore, on account of this ability and the proximity of cereal cultivation areas and vineyards to the saltern, it is reasonable to detect this yeast extensively throughout this habitat. Specifically, *S*. *bayanus*, detected in SE Spring, has been previously found to be a yeast associated with the winemaking process [40]. Other halotolerant fungi, such as *Cladosporium*, *Trimmatostroma*, and *Penicillum*, or the osmotolerant yeasts *Candida*, *Metschnikowia*, and *Pichia*, previously described in salty environments [7,14,15], have been detected in this continental saltern too. Several species of *Cladosporium* are considered true halophiles that are present in hypersaline environments of different geographical areas [41], and the presence of the halophilic *Trimmatostroma* has been associated with Adriatic salterns [42]. In addition, the genera *Metschnikowia*, *Hanseniaspora*, *Candida*, and *Taphrina* are typically associated with winemaking environments. In fact, *Metschnikowia* has been recovered from grapevine phyllospheres, fruit flies, grapes, and wine ferments, as part of the resident microbiota of wineries and winemaking equipment [43], while *Hanseniaspora* has been isolated in grape must, and in the winery environment [44]. Both *Candida* and *Taphrina* genera have been described as being associated with the grape berry microbial community, albeit as a minor percentage [45]. The presence of winemaking-related fungal genera could be the result of contamination via agriculture, possibly due to the proximity of significant cereal- and wine-producing areas of La Rioja, which borders the Añana Salt Valley. We may hypothesize that the proximity of vineyards, along with the relationship of the filtered rainwater to the spring water, could transport these fungi, which, in turn, end up in the colonization of the saltern, in addition to the fact that some of them have the ability to adapt to saline conditions. 

Among the *Basidiomycota* phylum, *Wallemia* was the only halophilic fungi detected in the saltern. This genus comprises one of the most xerophilic fungi ever described. Obligate halophilic species, such as *W*. *ichthyophaga* or *W*. *muriae*, have been described from brine of solar salterns [46]. In this study, *W*. *ichthyophaga* was detected in the brine. Other genera of the same phylum detected in our study were *Vishniacozyma*, *Peniophora*, *Stereum*, *Filobasidium*, and *Trametes*. Most of these are wood, plant, lichen, or fruit surface fungi distributed in natural environments. Specifically, *Vishniacozyma victoriae* and *Stereum hirsutum*, detected in this study, are related with lichens, mosses, and plants [47,48,49,50].

However, the high number of unassigned ASVs in this study, as has been observed in other natural environments [51,52], suggests that there are still fungal taxa to be discovered in this saltern. A large diversity of undescribed and unknown aquatic fungi, known as “dark matter fungi” [53], is usual in those environments [54], so a lack of taxonomy-associated sequences in databases could influence the low taxonomic assignment in metabarcoding-based environmental studies. 

Furthermore, not all rarefaction curves reached a plateau due to the limitation in sequencing depth. As low sample numbers were studied, it was considered preferable to include all sampling sites in the study despite the possible loss of diversity. The number of sequences obtained in the study might be influenced by the salt content of the water. It is well known that in HTS procedures, the presence of salt during the hybridization stage decreases specificity, favoring less stable bonds [55]. This results in lower quality sequences, which could be eliminated in the quality control process. Alpha diversity obtained in this study clearly showed one specific location where a lower fungal richness and diversity was found in comparison to the other locations that were examined. This location was SE Spring. Its Pielou’s evenness value (0.58) and Simpson index (0.77) were the lowest, probably due to the dominance of *Saccharomyces* (relative abundance of 49.9%), which as mentioned previously, could have ended up in the SE Spring as a result of rainwater filtration from adjacent agricultural fields. S8 piezometer groundwater also has low diversity (93 ASVs and a Shannon index of 4.89 versus other locations ranging from 262 to 396 ASVs and Shannon indexes of 5.44–6.47). It is important to consider that in regard to the S8 piezometer, the water source is protected from external influence, as it is underground. Thus, the arrival of fungal taxa not specific to the groundwater is more difficult and this may be the reason fewer taxa are detected there. The other locations of the saltern analyzed in this study had greater external influence; thereby, the detection of more fungal taxa and higher diversity could be explained as a result of closer contact with plants and animals (mainly arthropods). Therefore, for future studies, we need to consider differences in water origin, surrounding biotic factors, as well as human activities that could be associated with fungal diversity at each site. 

When the relation of water physicochemical parameters with diversity was studied, correlation between pH and fungal diversity was reported (*p*-value <0.05 for Shannon and Simpson indexes), showing that diversity decreases when pH decreases too. Liu et al. [56] also reported the same effect in soil samples, suggesting that pH could be an important predictor of fungal diversity. Bray–Curtis dissimilarity also showed that in this environment, fungi distribution is segregated in two different groups according to salinity: salty water and brackish water, statistically significantly dissimilar from each other. This phenomenon was also reported in a salinity gradient study on a marsh [57], suggesting that salinity influences distinctive fungal composition. Therefore, the diversity and composition results in this survey showed three differentiated water ecosystems in the saltern: the SE Spring, the two brackish-water sites, and the ponds and Pico Spring salty waters. This grouping accords with the clustering based on FUNGuild results, where these three groups were observed according to their trophic mode levels.

At SE Spring, the saprotrophic mode tends to be the most abundant life strategy. This result is supported by the dominance of *Saccharomyces* in this location, which is a well-known saprotroph. At Pico Spring and Ponds I, II, and III, the dominance of saprotrophs is also observed due to the presence of *Saccharomyces* and *Peniophora. Peniophora* acts as a saprotroph mainly on fallen branches and twigs but also on rotting trees [47]. The presence of saprotrophs is also due to *Wallemia* and *Filobasidium*, which are mainly plant-associated saprotrophs in saline and extreme environments [46,58]. Pico Spring is a natural spring that emanates directly from rocks in the mountains, surrounded by plants and soil. It is also interesting to consider that the structures of the channels that distribute the brine along the ponds in the saltern are constructed of wood. Therefore, the presence of wood-associated and plant-associated fungi could be enhanced in this environment. However, we could also see a considerable increment of pathotroph−saprotroph−symbiotrophs in Pond II and III due to the presence of *Epicoccum dendrobii*, which is a ubiquitous ascomycete. Some members of this genus are known to cause plant diseases, while others act as biological control agents against a range of plant pathogens [59]. The presence of the lichen parasite-lichenized guild in pathotroph-symbiotroph mode in Pond II and III is due to the abundance of the *Lecanoraceae* family. The pathotroph representation in these sampling sites is enhanced by the presence of the plant pathogen *Mycosphaerella* (probably *M*. *tassiana*) and the black yeast *Trimmatostroma salinum*. This last species was found growing in and on wood immersed in brine in another study [60]. It is thought that the presence of symbiont and saprotrophic fungi in saline environments might accelerate the leaching of dissolved organic matter (DOM) from the host, which then becomes available to other microorganisms in the marine environment [61], in this case, in the hypersaline one. Thus, the increment of fungal taxa associated with plants, lichens, and wood in these samples, especially in regard to Pond II and III, could be related to more direct environmental exposure and the possibility of obtaining substrates (of plant or animal origin) more easily in these sites than in others. 

Finally, in relation to Pico Dulce Spring and S8 piezometer, the pathotroph-saprotroph trophic mode was the main mode. This could be explained by the presence of *Ramularia*, which is considered a plant pathogenic hyphomycete [62], *Stemphylium*, whose members were isolated from wood specimens of a boat and demonstrated ligninase activity [63] and the genus *Didymella*, composed of opportunistic parasitic microorganisms that often take advantage of special conditions to settle on plants and occasionally cause serious damage [64]. Furthermore, we found an increase of pathotroph-saprotroph-symbiotrophs, especially the animal endosymbiont-animal pathogen-plant pathogen-undefined saprotroph guild, stemming from the abundance of *Vishniacozyma* and *Pichia* at both sampling sites. Both have been previously reported on grape berries [65]. Moreover, the higher relative abundance of plant pathogens was due to the presence of the genus *Mycosphaerella*, which as mentioned previously is a large and important genus of plant pathogens. 

These results showed the ubiquity of saprotrophs across the saltern. Our results demonstrated that the characteristics of the habitat determine the trophic guilds and the type of organisms present there. The fact that this continental saltern is surrounded by flora and fauna specific to the terrain and agricultural fields means that this environment presents mainly saprophytic organisms, with the presence of halotolerant and halophilic species among them. Furthermore, saprophytic fungi were more abundant in hypersaline water than in brackish water. In contrast, Yang and Sun [19] described a significantly lower abundance of plant saprotrophs and few saprotrophs in extremely saline soils in comparison with non-saline soils. 

## 5. Conclusions

This is the first study conducted on a continental-type solar saltern using metabarcoding sequencing technology based on ITS with the aim of examining fungal presence. It seems that this type of habitat is not solely influenced by the salinity itself. Other surrounding factors also have a direct or indirect influence regarding the distribution and appearance of fungal species, which in this case are distributed across at least three different aquatic systems. We sought to explore the reasons for the variation in the fungal composition among different locations in the valley, which needs to be further studied and examined with respect to additional environmental factors, such as rain, wind, vegetation, animal presence, or the relationship with the diapir. Nevertheless, the results contribute valuable information for increasing the understanding of how geographic location (mainly its relatedness to nearby vineyards and cereal fields), salinity, and environmental factors may play an important role in shaping the fungal presence in this saltern.

## Figures and Tables

**Figure 1 jof-07-01074-f001:**
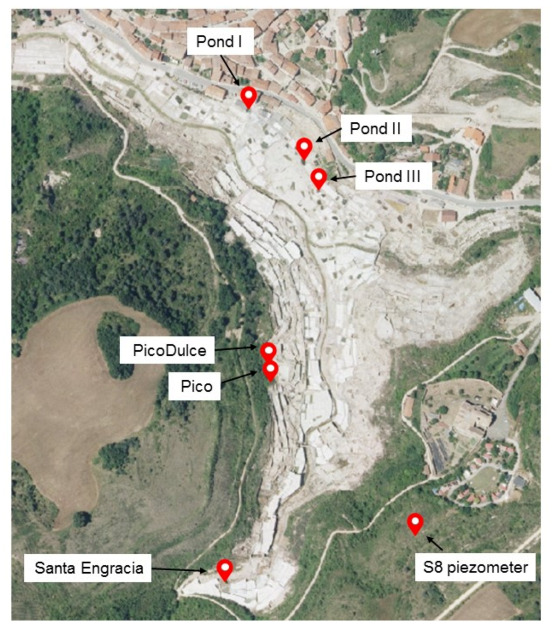
Locations along the Añana Salt Valley studied for fungal diversity and composition determination (photograph obtained from Visor geoEuskadi at https://www.geo.euskadi.eus/s69-bisorea/es/x72aGeoeuskadiWAR/index.jsp, accessed on 26 July 2021). The exact coordinates of each location are as follows: Pond I, 42.8014 N 2.9860 W; Pond II, 42.8009 N 2.9852 W; Pond III, 42.8006 N 2.9851 W; Pico Dulce, 42.7989 N 2.9858 W; Pico, 42.7986 N 2.9856 W; Santa Engracia, 42.7965 N 2.9863 W; S8 piezometer, 42.7970 N 2.9838 W.

**Figure 2 jof-07-01074-f002:**
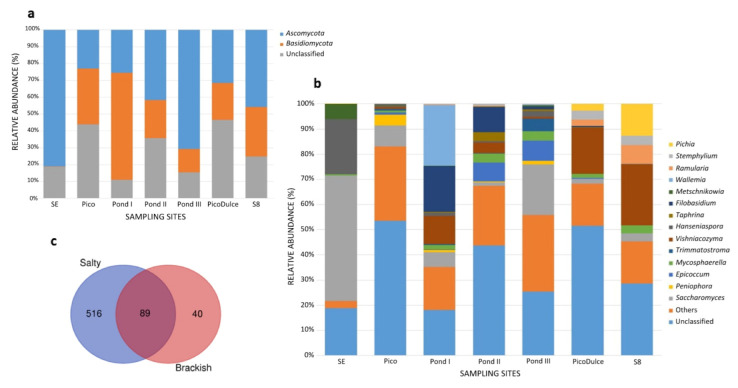
Relative abundance of the (**a**) phylum and (**b**) genera detected at the different locations analyzed in the saltern. Genera detected at <3% in all sampling sites are classified as Others; (**c**) Venn plot showing the share and exclusive genera between salty water location and brackish water location groups. SE, Santa Engracia.

**Figure 3 jof-07-01074-f003:**
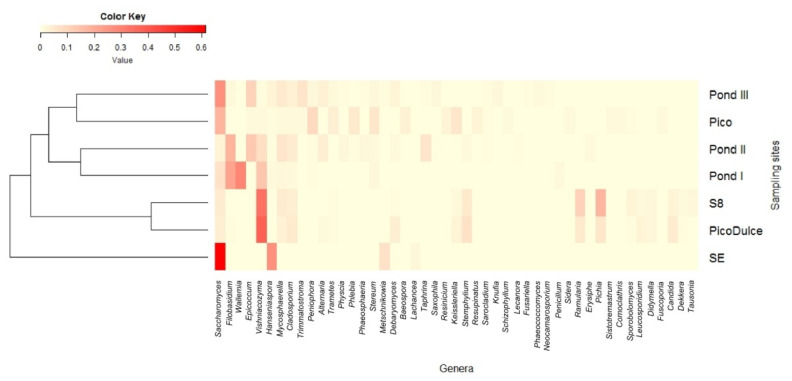
Heatmap showing the distribution of the most abundant genera along the sampling sites. The color represents the abundance of genera; the closer to red, the higher the abundance, and the closer to yellow, the lower the abundance. Clustering was performed based on the Bray–Curtis dissimilarity. SE, Santa Engracia.

**Figure 4 jof-07-01074-f004:**
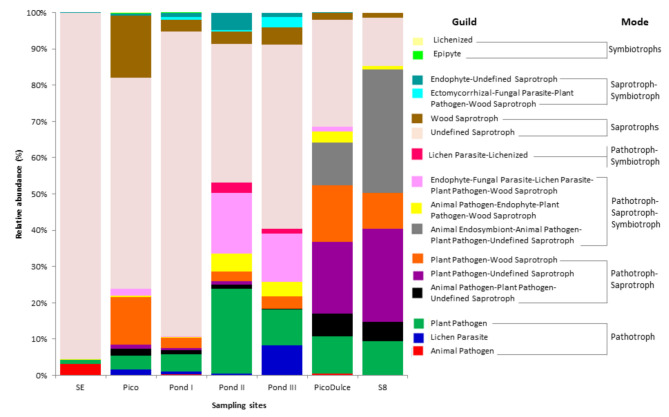
Relative abundance of the trophic modes and guilds detected at the sites analyzed. SE, Santa Engracia.

**Table 1 jof-07-01074-t001:** Alpha diversity indexes calculated for the locations of the saltern analyzed in the study.

Sample Site	Observed ASVs	Chao1	Shannon	Simpson	Pielou’s Evenness
Santa Engracia Spring	71	116	3.59	0.77	0.58
Pico Spring	354	633	6.27	0.95	0.74
Pond I	264	452	5.44	0.93	0.67
Pond II	396	699	6.47	0.94	0.75
Pond III	263	337	6.07	0.94	0.75
Pico Dulce Spring	262	430	6.15	0.96	0.76
S8	93	93	4.89	0.91	0.74

ASVs, amplicon sequence variants.

**Table 2 jof-07-01074-t002:** Correlation between water physicochemical parameters and fungal diversity. The analysis was carried out using Pearson’s coefficient test.

Parameter	Pearson’s Coefficient Test (*p* Values)	
	Shannon Index	Simpson Index
NaCl	0.896	0.732
Temperature	0.314	0.262
pH	0.029 *	0.015 *

* statistically significant correlation based on *p*-value (*p* < 0.05).

## Data Availability

The nucleotide sequence data from the study obtained after Illumina sequencing are available in the DDBJ/EMBL/GenBank databases under the accession number PRJNA749727. Other data presented in this study are available in Appendix A.

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
