# Peer review of "Fungal Diversity and Composition of the Continental Solar Saltern in Añana Salt Valley (Spain)"

_jof, 2021, doi:10.3390/jof7121074_

Round 1

Reviewer 1 Report

I believe this manuscript will be of interest for the reader of J. of Fungi. They look at seven sample sites for presence of Fungi in solar saltern environment in Spain. Fungi diversity were not well described in this area and new knowledge on this will be important to the literature. They use metagenomic with ITS sequencing with illumina sequencing. They presented a short study with only 7 sites but could be interesting for readers of the journal. They made some linkage about the trophic mode of different fungi and discuss relationship with culture or structure around, wood and agriculture to link the organisms found. My concern is more about they only made their diversity evaluation to the Genus level and not species. This would allow to target more specifically to the more specific trophic mode and would have been interesting to see species level. I can understand for some group resolution to specific species might not be enough but should be discussed in the manuscript about it and limitation. Some additional information related to some ASV and species. Also I believe the use of FUNGuild software will benefits to have more species ID. Other information may be need to improve the manuscript. Seem the info is in the Table S3 but why not evaluated for the species? In P5L183-184 you mention it but dont detail. Figure 2 only about Genera.

Specific comments:

P4 Section 3.2, for 7 samples only 704874 sequences seem quite low. I can understand diversity not as high, could you discuss about it in discussion? In your rarefaction curve you reach a plateau for S8 and SantaEngracia quickly but not for the other sites, could you discuss, you may not have sequenced deep enough.

P5L164-167: Where are those information, table graph? Where the information come from? Some info in Table S1 not indicated here but dont see the Pearson info there?

P6L198, Any information about sporulation of those fungi and potential viability? Epicoccum...

Interesting your discussion on saprotroph that represent the biggest proportion of the trophic.

Figure 4 P8, Same color repeat, not possible to differentiate which one is what, add more colors or texture. Any info on same graph to other Phylum, Species or family?

P9L290-293, interesting 50% not assigned ASV. Not to Family or other Phylum or just to species and genus level?

In Table 1 and in P9 you mention different value of diversity, could you explain more differences between them and at what level it is different, Pielou, Shannon, Simpson...

Reviewer 2 Report

  1. Line 164-167: How to determine the correlation between the physicochemical parameters of the water and the fungal diversity? Additionally, the results showed that the correlation of the diversity and pH was negative, but the number of samples in each sampling site was too small.
  2. Figure 1Recorded collection sites of the samples from the Añana Salt Valley. It's better to explain further sampling coordinates and times of all samples.
  3. Figures2, 3, 4, and S1 were not very clear and affect the result analysis.
  4. The results show that a negative correlation of the diversity with pH, and the differences in fungal diversity among groups based on their salinity, It should be discussed and compared with other literaturein the discussion.
  5. The resultsshould compare with the fungal diversity of other solar salterns in the discussion.
  6. The abbreviation of the name of the journal in the references should be unified, for example, Line 457 and 489.

Others:

Line 162: “S8 piezometer” in Table 1 and “S8” in Table S1 should be consistent.

Line 170: “Subterranean” in line 170 and “Groundwater” in Table S1 should be consistent.

Line 180: “manually” should be “manual”.

Reviewer 3 Report

This is an iteresting work, showing the "normal" yeast population in a saline environment. I have only one doubt: ¿Being S. cerevisiae the mot frequent yeast isolated, does it have different capacities to grow and function in saline media? I think this should be considered s an interesting matter to study in the future.

Reviewer 4 Report

Manuscript title “Fungal diversity and composition of the continental solar saltern in Añana Salt Valley (Spain)”

Authored by Maia Azpiazu-Muniozguren et al.

Comments:

The manuscript is well organized with respect to proposed study, planning to execute study, continuation of the manuscript and English language. However, authors need to address some queries given as below:

  1. The Direct DNA isolation of fungal samples from water samples collected from various study sites and their sequencing is now a developing as culture independent technique used for fungal samples from various sources. However, the culture based basic methods of fungal identification are still equally important to study fungal diversity of any substrate or any locality. Therefore, authors are advised to don’t ignore this aspect of morpho-taxonomy of fungi and add this portion in materials and methods as well as in result section, if possible.
  2. Add reference for Physicochemical parameters analyses of water samples.
  3. The scientific names of fungal genera in figure 3 should be in italics.
  4. Add some microscopic photographs of fungi, if possible.
  • Correct Line 72-72 as:

One water sample from a total of seven places were studied (Figure 1) in this survey

Change to

One water sample from all seven places (Figure 1) were collected during the survey and studied for fungal diversity.

  • Check bibliographic references with text citations once.

Round 2

Reviewer 2 Report

Dear Editor and authors,

Thank you very much for taking my request into consideration, but I am still unable to endorse the publication. Please find my comments below:

  1. "We agree in that the number of samples is perhaps too small to be conclusive. "

-> The results showed that the diversity and pH showed a statistically significant correlation. However, ITS metaarcoding was measured only once for each sample. So, I still think that the number of samples in each sampling site was too small.

  1. "The time of sampling has been added in Materials and Methods 2.1 section."

-> I have one question about this response: The sampling time interval of different locations along the saltern was about eight months. Does time affect fungal diversity of the sample? In addition, if it is the sampling time in the manuscript, what is the significance of comparing fungal diversity in samples from different locations at different times?

Finally, I think some issues need clarification or further analysis.
